# Monotonic Multihead Attention

**Xutai Ma[2] \*, Juan Pino[1], James Cross[1], Liezl Puzon[1], Jiatao Gu[1]**

[1]Facebook
[2]Johns Hopkins University

`xutai_ma@jhu.edu, puzon@cs.stanford.edu`
`{juancarabina,jcross,jgu}@fb.com`

## Abstract

Simultaneous machine translation models start generating a target sequence before they have encoded the source sequence. Recent approaches for this task either apply a fixed policy on a state-of-the art Transformer model, or a learnable monotonic attention on a weaker recurrent neural network-based structure. In this paper, we propose a new attention mechanism, Monotonic Multihead Attention (MMA), which extends the monotonic attention mechanism to multihead attention. We also introduce two novel and interpretable approaches for latency control that are specifically designed for multiple attention heads. We apply MMA to the simultaneous machine translation task and demonstrate better latency-quality tradeoffs compared to MILk, the previous state-of-the-art approach. We analyze how the latency controls affect the attention span and we study the relationship between the speed of a head and the layer it belongs to. Finally, we motivate the introduction of our model by analyzing the effect of the number of decoder layers and heads on quality and latency.[1]

## 1 Introduction

Simultaneous machine translation adds the capability of a live interpreter to machine translation: a simultaneous model starts generating a translation before it has finished reading the entire source sentence. Such models are useful in any situation where translation needs to be done in real time. For example, simultaneous models can translate live video captions or facilitate conversations between people speaking different languages. In a usual translation model, the encoder first reads the entire sentence, then the decoder writes the target sentence. On the other hand, a simultaneous neural machine translation model alternates between reading the input and writing the output using either a fixed or learned policy.

Monotonic attention mechanisms fall into the flexible policy category, in which the policies are automatically learned from data. Recent work exploring monotonic attention variants for simultaneous translation include: hard monotonic attention (Raffel et al., 2017), monotonic chunkwise attention (MoChA) (Chiu & Raffel, 2018) and monotonic infinite lookback attention (MILk) (Arivazhagan et al., 2019). MILk in particular has shown better quality/latency trade-offs than fixed policy approaches, such as wait-$k$ (Ma et al., 2019) or wait-if-* (Cho & Esipova, 2016) policies. MILk also outperforms hard monotonic attention and MoChA; while the other two monotonic attention mechanisms only consider a fixed window, MILk computes a softmax attention over all previous encoder states, which may be the key to its improved latency-quality tradeoffs. These monotonic attention approaches also provide a closed-form expression for the expected alignment between source and target tokens.

However, monotonic attention-based models, including the state-of-the-art MILk, were built on top of RNN-based models. RNN-based models have been outperformed by the recent state-of-the-art Transformer model (Vaswani et al., 2017), which features multiple encoder-decoder attention layers and multihead attention at each layer.

---

\*Work conducted during an internship at Facebook

[1]The code is available at `https://github.com/pytorch/fairseq/tree/master/examples/simultaneous_translation`

We thus propose monotonic multihead attention (MMA), which combines the high translation quality from multilayer multihead attention and low latency from monotonic attention. We propose two variants, Hard MMA (MMA-H) and Infinite Lookback MMA (MMA-IL). MMA-H is designed with streaming systems in mind where the attention span must be limited. MMA-IL emphasizes the quality of the translation system. We also propose two novel latency regularization methods. The first encourages the model to be faster by directly minimizing the average latency. The second encourages the attention heads to maintain similar positions, preventing the latency from being dominated by a single or a few heads.

The main contributions of this paper are: (1) A novel monotonic attention mechanism, monotonic multihead attention, which enables the Transformer model to perform online decoding. This model leverages the power of the Transformer and the efficiency of monotonic attention. (2) Better latency/quality tradeoffs compared to the MILk model, the previous state-of-the-art, on two standard translation benchmarks, IWSLT15 English-Vietnamese (En-Vi) and WMT15 German-English (De-En). (3) Analyses on how our model is able to control the attention span and on the relationship between the speed of a head and the layer it belongs to. We motivate the design of our model with an ablation study on the number of decoder layers and the number of decoder heads.

## 2 MONOTONIC MULTIHEAD ATTENTION MODEL

In this section, we review the monotonic attention-based approaches in RNN-based encoder-decoder models. We then introduce the two types of Monotonic Multihead Attention (MMA) for Transformer models: MMA-H and MMA-IL. Finally, we introduce strategies to control latency and coverage.

### 2.1 MONOTONIC ATTENTION

The hard monotonic attention mechanism (Raffel et al., 2017) was first introduced in order to achieve online linear time decoding for RNN-based encoder-decoder models. We denote the input sequence as $\mathbf{x} = \{x_1, ..., x_T\}$, and the corresponding encoder states as $\mathbf{m} = \{m_1, ..., m_T\}$, with $T$ being the length of the source sequence. The model generates a target sequence $\mathbf{y} = \{y_1, ..., y_U\}$ with $U$ being the length of the target sequence. At the $i$-th decoding step, the decoder only attends to one encoder state $m_{t_i}$ with $t_i = j$. When generating a new target token $y_i$, the decoder chooses whether to move one step forward or to stay at the current position based on a Bernoulli selection probability $p_{i,j}$, so that $t_i \geq t_{i-1}$. Denoting the decoder state at the $i$-th position, starting from $j = t_{i-1}, t_{i-1} + 1, t_{i-1} + 2, ...$, this process can be calculated as follows: [2]

$$e_{i,j} = \text{MonotonicEnergy}(s_{i-1}, m_j) \tag{1}$$

$$p_{i,j} = \text{Sigmoid}(e_{i,j}) \tag{2}$$

$$z_{i,j} \sim \text{Bernoulli}(p_{i,j}) \tag{3}$$

When $z_{i,j} = 1$, we set $t_i = j$ and start generating a target token $y_i$; otherwise, we set $t_i = j + 1$ and repeat the process. During training, an expected alignment $\boldsymbol{\alpha}$ is introduced to replace the softmax attention. It can be calculated in a recurrent manner, shown in Equation 4:

$$\alpha_{i,j} = p_{i,j} \sum_{k=1}^{j} \left( \alpha_{i-1,k} \prod_{l=k}^{j-1} (1 - p_{i,l}) \right)$$
$$= p_{i,j} \left( (1 - p_{i,j-1}) \frac{\alpha_{i,j-1}}{p_{i,j-1}} + \alpha_{i-1,j} \right) \tag{4}$$

Raffel et al. (2017) also introduce a closed-form parallel solution for the recurrence relation in Equation 5:

$$\alpha_{i,:} = p_{i,:} \text{cumprod}(1 - p_{i,:}) \text{cumsum} \left( \frac{\alpha_{i-1,:}}{\text{cumprod}(1 - p_{i,:})} \right) \tag{5}$$

where $\text{cumprod}(\boldsymbol{x}) = [1, x_1, x_1 x_2, ..., \prod_{i=1}^{|\boldsymbol{x}|-1} x_i]$ and $\text{cumsum}(\boldsymbol{x}) = [x_1, x_1 + x_2, ..., \sum_{i=1}^{|\boldsymbol{x}|} x_i]$. In practice, the denominator in Equation 5 is clamped into a range of $[\epsilon, 1]$ to avoid numerical instabilities introduced by $\text{cumprod}$. Although this monotonic attention mechanism achieves online

---

[2]Note that during training, to encourage discreteness, Raffel et al. (2017) added a zero mean, unit variance pre-sigmoid noise to $e_{i,j}$.

linear time decoding, the decoder can only attend to one encoder state. This limitation can diminish translation quality as there may be insufficient information for reordering.

Moreover, the model lacks a mechanism to adjust latency based on different requirements at decoding time. To address these issues, Chiu & Raffel (2018) introduce Monotonic Chunkwise Attention (MoChA), which allows the decoder to apply softmax attention to a fixed-length subsequence of encoder states. Alternatively, Arivazhagan et al. (2019) introduce Monotonic Infinite Lookback Attention (MILk) which allows the decoder to access encoder states from the beginning of the source sequence. The expected attention for the MILk model is defined in Equation 6.

$$\beta_{i,j} = \sum_{k=j}^{|\boldsymbol{x}|} \left( \frac{\alpha_{i,k} \exp(u_{i,j})}{\sum_{l=1}^{k} \exp(u_{i,l})} \right) \tag{6}$$

## 2.2 Monotonic Multihead Attention

Previous monotonic attention approaches are based on RNN encoder-decoder models with a single attention and haven't explored the power of the Transformer model. [3] The Transformer architecture (Vaswani et al., 2017) has recently become the state-of-the-art for machine translation (Barrault et al., 2019). An important feature of the Transformer is the use of a separate multihead attention module at each layer. Thus, we propose a new approach, Monotonic Multihead Attention (MMA), which combines the expressive power of multihead attention and the low latency of monotonic attention.

Multihead attention allows each decoder layer to have multiple heads, where each head can compute a different attention distribution. Given queries $Q$, keys $K$ and values $V$, multihead attention MultiHead$(Q, K, V)$ is defined in Equation 7.

$$\text{MultiHead}(Q, K, V) = \text{Concat}(\text{head}_1, ..., \text{head}_H)W^O$$
$$\text{where } \text{head}_h = \text{Attention}\left(QW_h^Q, KW_h^K, VW_h^V, \right) \tag{7}$$

The attention function is the scaled dot-product attention, defined in Equation 8:

$$\text{Attention}(Q, K, V) = \text{Softmax}\left( \frac{QK^T}{\sqrt{d_k}} \right) V \tag{8}$$

There are three applications of multihead attention in the Transformer model:

1. The **Encoder** contains self-attention layers where all of the queries, keys and values come from previous layers.
2. The **Decoder** contains self-attention layers that allow each position in the decoder to attend to all positions in the decoder up to and including that position.
3. The **Encoder-Decoder attention** contains multihead attention layers where queries come from the previous decoder layer and the keys and values come from the output of the encoder. Every decoder layer has a separate encoder-decoder attention.

For MMA, we assign each head to operate as a separate monotonic attention in encoder-decoder attention.

For a transformer with $L$ decoder layers and $H$ attention heads per layer, we define the selection process of the $h$-th head encoder-decoder attention in the $l$-th decoder layer as

$$e_{i,j}^{l,h} = \left( \frac{m_j W_{l,h}^K (s_{i-1} W_{l,h}^Q)^T}{\sqrt{d_k}} \right)_{i,j} \tag{9}$$

$$p_{i,j}^{l,h} = \text{Sigmoid}(e_{i,j}) \tag{10}$$

$$z_{i,j}^{l,h} \sim \text{Bernoulli}(p_{i,j}) \tag{11}$$

---

[3]MILk was based on a strengthened RNN-based model called RNMT+. The original RNMT+ model (Chen et al., 2018) uses multihead attention, computes attention only once, and then concatenates that single attention layer to the output of each decoder layer block. However, the RNMT+ model used for MILk in Arivazhagan et al. (2019) only uses a single head.

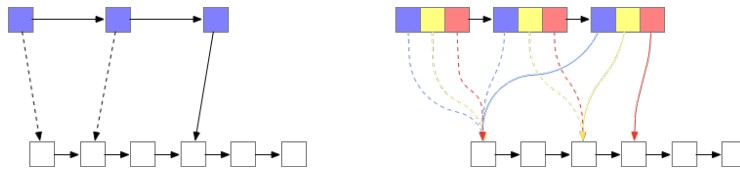

**Figure 1:** Monotonic Attention (Left) versus Monotonic Multihead Attention (Right).

where $W_{l,h}$ is the input projection matrix, $d_k$ is the dimension of the attention head. We make the selection process independent for each head in each layer. We then investigate two types of MMA, MMA-H(ard) and MMA-IL(infinite lookback). For MMA-H, we use Equation 4 in order to calculate the expected alignment for each layer each head, given $p_{i,j}^{l,h}$. For MMA-IL, we calculate the softmax energy for each head as follows:

$$u_{i,j}^{l,h} = \text{SoftEnergy} = \left( \frac{m_j \hat{W}_{l,h}^K (s_{i-1} \hat{W}_{l,h}^Q)^T}{\sqrt{d_k}} \right)_{i,j} \tag{12}$$

and then use Equation 6 to calculate the expected attention. Each attention head in MMA-H hard-attends to one encoder state. On the other hand, each attention head in MMA-IL can attend to all previous encoder states. Thus, MMA-IL allows the model to leverage more information for translation, but MMA-H may be better suited for streaming systems with stricter efficiency requirements. Finally, our models use unidirectional encoders: the encoder self-attention can only attend to previous states, which is also required for simultaneous translation.

At inference time, our decoding strategy is shown in Algorithm 1. For each $l, h$, at decoding step $i$, we apply the sampling processes discussed in subsection 2.1 individually and set the encoder step at $t_i^{l,h}$. Then a hard alignment or partial softmax attention from encoder states, shown in Equation 13, will be retrieved to feed into the decoder to generate the $i$-th token. The model will write a new target token only after all the attentions have decided to write. In other words, the heads that have decided to write must wait until the others have finished reading.

$$c_i^l = \text{Concat}(c_i^{l,1}, c_i^{l,2}, ..., c_i^{l,H})$$

$$\text{where } c_i^{l,h} = f_{\text{context}}(\boldsymbol{h}, t_i^{l,h}) = \begin{cases} m_{t_i^{l,h}} & \text{MMA-H} \\ \sum_{j=1}^{t_i^{l,h}} \frac{\exp\left(u_{i,j}^{l,h}\right)}{\sum_{j=1}^{t_i^{l,h}} \exp\left(u_{i,j}^{l,h}\right)} m_j & \text{MMA-IL} \end{cases} \tag{13}$$

Figure 1 illustrates a comparison between our model and the monotonic model with one attention head. Compared with the monotonic model, the MMA model is able to set attention to different positions so that it can still attend to previous states while reading each new token. Each head can adjust its speed on-the-fly. Some heads read new inputs, while the others can stay in the past to retain the source history information. Even with the hard alignment variant (MMA-H), the model is still able to preserve the history information by setting heads to past states. In contrast, the hard monotonic model, which only has one head, loses the previous information at the attention layer.

## 2.3 LATENCY CONTROL

Effective simultaneous machine translation must balance quality and latency. At a high level, latency measures how many source tokens the model has read until a translation is generated. The model we have introduced in subsection 2.2 is not able to control latency on its own. While MMA allows simultaneous translation by having a read or write schedule for each head, the overall latency is determined by the fastest head, i.e. the head that reads the most. It is possible that a head always reads new input without producing output, which would result in the maximum possible latency. Note that the attention behaviors in MMA-H and MMA-IL can be different. In MMA-IL, a head reaching the end of the sentence will provide the model with maximum information about the source sentence. On the other hand, in the case of MMA-H, reaching the end of sentence for a head only

---

**Algorithm 1** MMA monotonic decoding. Because each head is independent, we compute line 3 to 16 in parallel

---

**Input:** $\boldsymbol{x}$ = source tokens, $\boldsymbol{h}$ = encoder states, $i = 1, j = 1, t_0^{l,h} = 1, y_0$ = StartOfSequence.
1: **while** $y_{i-1} \neq$ EndOfSequence **do**
2:     $t_{\max} = 1$
3:     $\boldsymbol{h}$ = empty sequence
4:     **for** $l \leftarrow 1$ to $L$ **do**
5:        **for** $h \leftarrow 1$ to $H$ **do**
6:           **for** $j \leftarrow t_{i-1}^{l,h}$ to $|\boldsymbol{x}|$ **do**
7:             $p_{i,j}^{l,h} = $ Sigmoid $\left(\text{MonotonicEnergy}(s_{i-1,m_j})\right)$
8:             **if** $p_{i,j}^{l,h} > 0.5$ **then**
9:                $t_i^{l,h} = j$
10:                $c_i^{l,h} = f_{\text{context}}(\boldsymbol{h}, t_i^{l,h})$
11:                **Break**
12:             **else**
13:                **if** $j > t_{\max}$ **then**
14:                    Read token $x_j$
15:                    Calculate state $h_j$ and append to $\boldsymbol{h}$
16:                    $t_{\max} = j$
17:           $c_i^l = \text{Concat}(c_i^{l,1}, c_i^{l,2}, ..., c_i^{l,H})$
18:           $s_i^l = \text{DecoderLayer}^l(s_{1:i-1}^l, s_{1:i-1}^{l-1}, c_i^l)$
19:     $y_i = \text{Output}(s_i^L)$
20:     $i = i + 1$

---

gives a hard alignment to the end-of-sentence token, which provides very little information to the decoder. Furthermore, it is possible that an MMA-H attention head stays at the beginning of sentence without moving forward. Such a head would not cause latency issues but would degrade the model quality since the decoder would not have any information about the input. In addition, this behavior is not suited for streaming systems.

To address these issues, we introduce two latency control methods. The first one is *weighted* average latency, shown in Equation 14:

$$g_i^W = \frac{\exp(g_i^{l,h})}{\sum_{l=1}^{L} \sum_{h=1}^{H} \exp(g_i^{l,h})} g_i^{l,h} \tag{14}$$

where $g_i^{l,h} = \sum_{j=1}^{|\boldsymbol{x}|} j \alpha_{i,j}$. Then we calculate the latency loss with a differentiable latency metric $\mathcal{C}$.

$$L_{avg} = \mathcal{C}\left(\boldsymbol{g}^W\right) \tag{15}$$

Like Arivazhagan et al. (2019), we use the Differentiable Average Lagging. It is important to note that, unlike the original latency augmented training in Arivazhagan et al. (2019), Equation 15 is not the expected latency metric given $\mathcal{C}$, but weighted average $\mathcal{C}$ on all the attentions. The real expected latency is $\hat{\boldsymbol{g}} = \max_{l,h}\left(\boldsymbol{g}^{l,h}\right)$ instead of $\bar{\boldsymbol{g}}$, but using this directly would only affect the speed of the fastest head. Equation 15 can control every head in a way that the faster heads will be automatically assigned to larger weights and slower heads will also be moderately regularized. For MMA-H models, we found that the latency of are mainly due to outliers that skip almost every token. The weighted average latency loss is not sufficient to control the outliers. We therefore introduce the head divergence loss, the average variance of expected delays at each step, defined in Equation 16:

$$L_{var} = \frac{1}{LH} \sum_{l=1}^{L} \sum_{h=1}^{H} \left(g_i^{l,h} - \bar{g}_i\right)^2 \tag{16}$$

where $\bar{g}_i = \frac{1}{LH} \sum g_i$ The final objective function is presented in Equation 17:

$$L(\theta) = -\log(\boldsymbol{y} \mid \boldsymbol{x}; \theta) + \lambda_{avg} L_{avg} + \lambda_{var} L_{var} \tag{17}$$

where $\lambda_{avg}, \lambda_{var}$ are hyperparameters that control both losses. Intuitively, while $\lambda_{avg}$ controls the overall speed, $\lambda_{var}$ controls the divergence of the heads. Combining these two losses, we are able to dynamically control the range of attention heads so that we can control the latency and the reading buffer. For MMA-IL model, we only use $L_{avg}$; for MMA-H we only use $L_{var}$.

## 3 EXPERIMENTAL SETUP

### 3.1 EVALUATION METRICS

We evaluate our model using quality and latency. For translation quality, we use tokenized BLEU [4] for IWSLT15 En-Vi and detokenized BLEU with SacreBLEU (Post, 2018) for WMT15 De-En. For latency, we use three different recent metrics, **Average Proportion** (AP) (Cho & Esipova, 2016), **Average Lagging** (AL) (Ma et al., 2019) and **Differentiable Average Lagging** (DAL) (Arivazhagan et al., 2019) [5]. We remind the reader of the metric definitions in Appendix A.2.

### 3.2 DATASETS

| Dataset | Train | Validation | Test |
|---|---|---|---|
| IWSLT15 En-Vi | 133k | 1268 | 1553 |
| WMT15 De-En | 4.5M | 3000 | 2169 |

**Table 1:** Number of sentences in each split.

| Dataset | RNN | Transformer |
|---|---|---|
| IWSLT15 En-Vi | 25.6 [6] | 28.7 |
| WMT15 De-En | 28.4 (Arivazhagan et al., 2019) | 32.3 |

**Table 2:** Offline model performance with unidirectional encoder and greedy decoding.

| Dataset | Beam Search | Bidirectional Encoder | Unidirectional Encoder |
|---|---|---|---|
| WMT15 De-En | 1 | 32.6 | 32.3 |
| | 4 | 33.0 | 33.0 |
| IWSLT15 En-Vi | 1 | 28.7 | 29.4 |
| | 10 | 28.8 | 29.5 |

**Table 3:** Effect of using a unidirectional encoder and greedy decoding to BLEU score.

We evaluate our method on two standard machine translation datasets, IWSLT14 En-Vi and WMT15 De-En. Statistics of the datasets can be found in Table 1. For each dataset, we apply tokenization with the Moses (Koehn et al., 2007) tokenizer and preserve casing.

**IWSLT15 English-Vietnamese** TED talks from IWSLT 2015 Evaluation Campaign (Cettolo et al., 2016). We follow the settings from Luong & Manning (2015) and Raffel et al. (2017). We replace words with frequency less than 5 by <*unk*>. We use tst2012 as a validation set tst2013 as a test set.

**WMT15 German-English** We follow the setting from Arivazhagan et al. (2019). We apply byte pair encoding (BPE) (Sennrich et al., 2016) jointly on the source and target to construct a shared vocabulary with 32K symbols. We use newstest2013 as validation set and newstest2015 as test set.

### 3.3 MODELS

We evaluate MMA-H and MMA-IL models on both datasets. The MILK model we evaluate on IWSLT15 En-Vi is based on Luong et al. (2015) rather than RNMT+ (Chen et al., 2018). In general, our offline models use unidirectional encoders, i.e. the encoder self-attention can only attend to previous states, and greedy decoding. We report offline model performance in Table 2 and the effect of using unidirectional encoders and greedy decoding in Table 3. For MMA models, we replace the encoder-decoder layers with MMA and keep other hyperparameter settings the same as the offline model. Detailed hyperparameter settings can be found in subsection A.1. We use the Fairseq library (Ott et al., 2019) for our implementation.

---

[4]We acquire the data from https://nlp.stanford.edu/projects/nmt/, which is tokenized. We do not have the tokenizer which processed this data, thus we report tokenize d BLEU for IWSLT15

[5]Latency metrics are computed on BPE tokens for WMT15 De-En – consistent with Arivazhagan et al. (2019) – and on word tokens for IWSLT15 En-Vi.

[6] Luong & Manning (2015) report a BLEU score of 23.0 but they didn't mention what type of BLEU score they used. This score is from our implementation on the data aquired from https://nlp.stanford.edu/projects/nmt/

## 4  RESULTS

In this section, we present the main results of our model in terms of latency-quality tradeoffs, ablation studies and analyses. In the first study, we analyze the effect of the variance loss on the attention span. Then, we study the effect of the number of decoder layers and decoder heads on quality and latency. We also provide a case study for the behavior of attention heads in an example. Finally, we study the relationship between the rank of an attention head and the layer it belongs to.

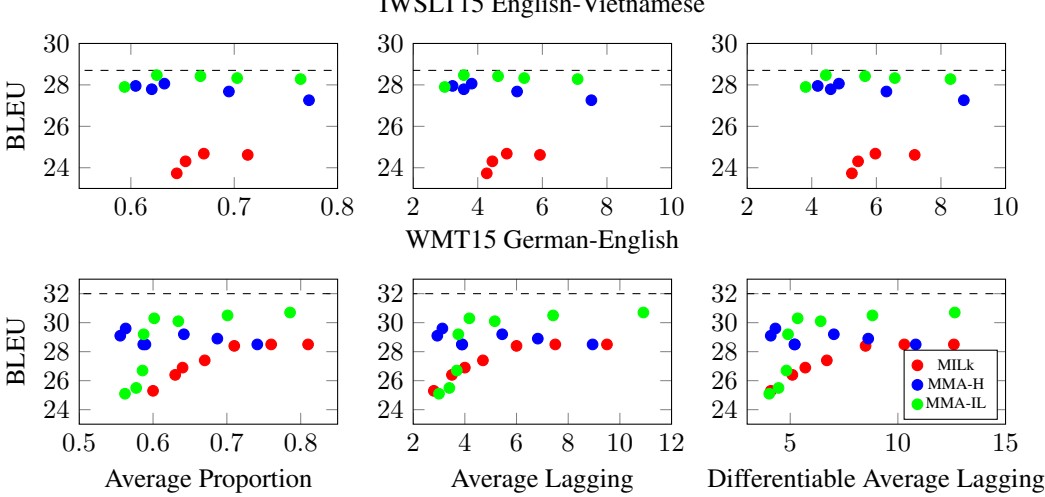

**Figure 2:** Latency-quality tradeoffs for MILk (Arivazhagan et al., 2019) and MMA on IWSLT15 En-Vi and WMT15 De-En. Black dashed line indicates the unidirectional offline transformer model with greedy search.

### 4.1  LATENCY-QUALITY TRADEOFFS

We plot the quality-latency curves for MMA-H and MMA-IL in Figure 2. The BLEU and latency scores on the test sets are generated by setting a latency range and selecting the checkpoint with best BLEU score on the validation set. We use differentiable average lagging (Arivazhagan et al., 2019) when setting the latency range. We find that for a given latency, our models obtain a better translation quality. While MMA-IL tends to have a decrease in quality as the latency decreases, MMA-H has a small gain in quality as latency decreases: a larger latency does not necessarily mean an increase in source information available to the model. In fact, the large latency is from the outlier attention heads, which skip the entire source sentence and point to the end of the sentence. The outliers not only increase the latency but they also do not provide useful information. We introduce the attention variance loss to eliminate the outliers, as such a loss makes the attention heads focus on the current context for translating the new target token.

It is interesting to observe that MMA-H has a better latency-quality tradeoff than MILk[7] even though each head only attends to only one state. Although MMA-H is not yet able to handle an arbitrarily long input (without resorting to segmenting the input), since both encoder and decoder self-attention have an infinite lookback, that model represents a good step in that direction.

### 4.2  ATTENTION SPAN

In subsection 2.3, we introduced the attention variance loss to MMA-H in order to prevent outlier attention heads from increasing the latency or increasing the attention span. We have already evaluated the effectiveness of this method on latency in subsection 4.1. We also want to measure the difference between the fastest and slowest heads at each decoding step. We define the average

---

[7]The numbers of MILk on WMT15 De-En are from Arivazhagan et al. (2019)

attention span in Equation 18:

$$\bar{S} = \frac{1}{|\boldsymbol{y}|} \left( \sum_i^{|\boldsymbol{y}|} \max_{l,h} t_i^{l,h} - \min_{l,h} t_i^{l,h} \right) \tag{18}$$

It estimates the reading buffer we need for streaming translation. We show the relation between the average attention span versus $\lambda_{var}$ in Figure 3. As expected, the average attention span is reduced as we increase $\lambda_{var}$.

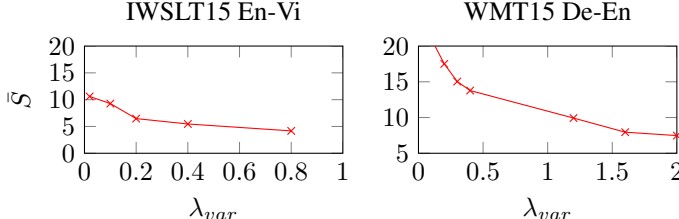

**Figure 3:** Effect of $\lambda_{var}$ on the average attention span. The variance loss works as intended by reducing the span with higher weights.

### 4.3 EFFECT ON NUMBER OF LAYERS AND NUMBER OF HEADS

One motivation to introduce MMA is to adapt the Transformer, which is the current state-of-the-art model for machine translation, to online decoding. Important features of the Transformer architecture include having a separate attention layer for each decoder layer block and multihead attention. In this section, we test the effect of these two components on the offline, MMA-H, and MMA-IL models from a quality and latency perspective. We report quality as measured by detokenized BLEU and latency as measured by DAL on the WMT13 validation set in Figure 4. We set $\lambda_{avg} = 0.2$ for MMA-IL and $\lambda_{var} = 0.2$ for MMA-H.

The offline model benefits from having more than one decoder layer. In the case of 1 decoder layer, increasing the number of attention heads is beneficial but in the case of 3 and 6 decoder layers, we do not see much benefit from using more than 2 heads. The best performance is obtained for 3 layers and 2 heads (6 effective heads). The MMA-IL model behaves similarly to the offline model, and the best performance is observed with 6 layers and 4 heads (24 effective heads). For MMA-H, with 1 layer, performance improves with more heads. With 3 layers, the single-head setting is the most effective (3 effective heads). Finally, with 6 layers, the best performance is reached with 16 heads (96 effective heads).

The general trend we observe is that performance improves as we increase the number of effective heads, either from multiple layers or multihead attention, up to a certain point, then either plateaus or degrades. This motivates the introduction of the MMA model.

We also note that latency increases with the number of effective attention heads. This is due to having fixed loss weights: when more heads are involved, we should increase $\lambda_{var}$ or $\lambda_{avg}$ to better control latency.

### 4.4 ATTENTION BEHAVIORS

We characterize attention behaviors by providing a running example of MMA-H and MMA-IL, shown in Figure 5. Each curve represents the path that an attention head goes through at inference time. For MMA-H, shown in Figure 5a, we found that when the source and target tokens have the same order, the attention heads behave linearly and the distance between fastest head and slowest head is small. For example, this can be observed from partial sentence pair "I also didn't know that" and target tokens "Tôi cũng không biết rằng", which have the same order. However, when the source tokens and target tokens have different orders, such as "the second step" and "bước (step) thứ hai (second)", the model will generate "bước (step)" first and some heads will stay in the past to retain

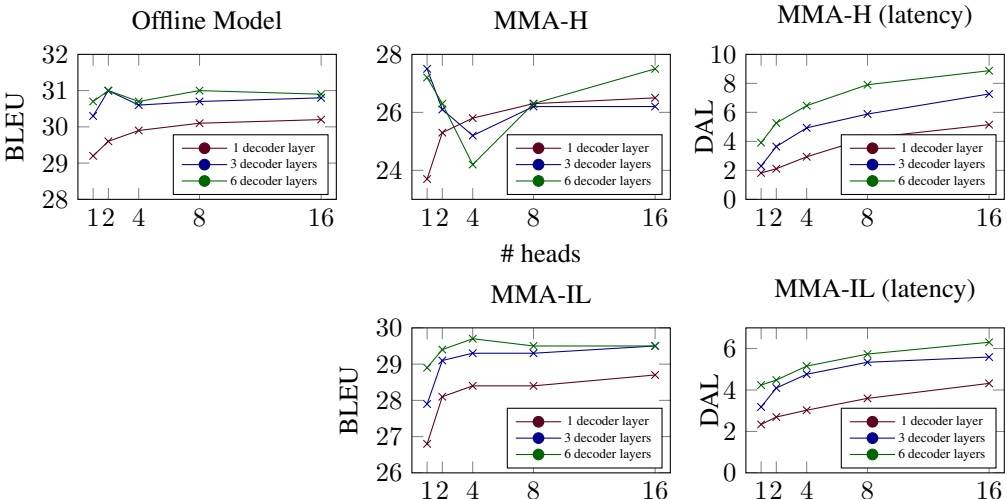

**Figure 4:** Effect of the number of decoder attention heads and the number of decoder attention layers on quality and latency, reported on the WMT13 validation set.

the information for later reordered translation "thứ hai (*second*)". We can also see that the attention heads have a near-diagonal trajectory, which is appropriate for streaming inputs.

The behavior of the heads in MMA-IL models is shown in Figure 5b. Notice that we remove the partial softmax alignment in this figure. We don't expect streaming capability for MMA-IL: some heads stop at early position of the source sentence to retain the history information. Moreover, because MMA-IL has more information when generating a new target token, it tends to produce translations with better quality. In this example, the MMA-IL model has a better translation on "isolate the victim" than MMA-H ("là cô lập nạn nhân" vs "là tách biệt nạn nhân")

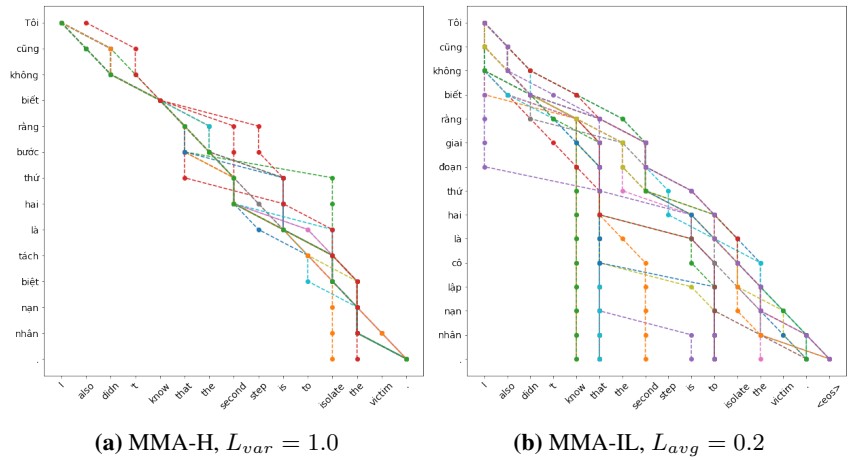

**(a)** MMA-H, $L_{var} = 1.0$        **(b)** MMA-IL, $L_{avg} = 0.2$

**Figure 5:** Running examples on IWSLT15 English-Vietnamese dataset

## 4.5 RANK OF THE HEADS

In Figure 6, we calculate the average and standard deviation of rank of each head when generating every target token. For MMA-IL, we find that heads in lower layers tend to have higher rank and are thus slower. However, in MMA-H, the difference of the average rank are smaller. Furthermore, the standard deviation is very large which means that the order of the heads in MMA-H changes frequently over the inference process.

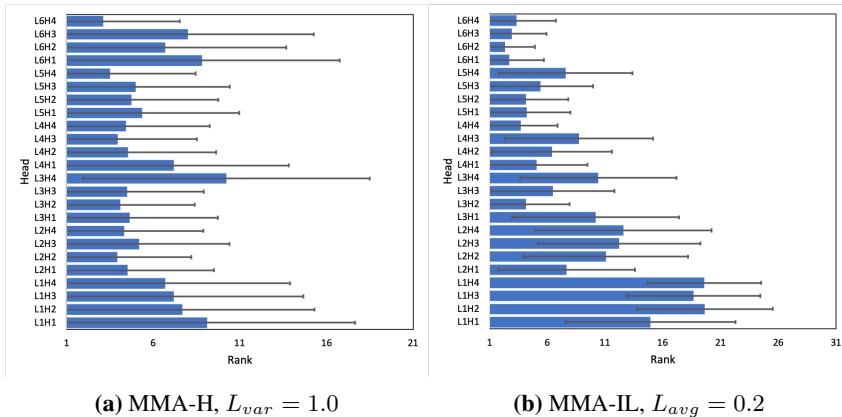

**(a)** MMA-H, $L_{var} = 1.0$        **(b)** MMA-IL, $L_{avg} = 0.2$

**Figure 6:** The average rank of attention heads during inference on IWSLT15 En-Vi. Error bars indicate the standard deviation. L indicates the layer number and H indicates the head number.

## 5  RELATED WORK

Recent work on simultaneous machine translation falls into three categories. In the first one, models use a rule-based policy for reading input and writing output. Cho & Esipova (2016) propose a Wait-If-* policy to enable an offline model to decode simultaneously. Ma et al. (2019) propose a wait-$k$ policy where the model first reads $k$ tokens, then alternates between read and write actions. Dalvi et al. (2018) propose an incremental decoding method, also based on a rule-based schedule. In the second category, a flexible policy is learnt from data. Grissom II et al. (2014) introduce a Markov chain to phrase-based machine translation models for simultaneous machine translation, in which they apply reinforcement learning to learn the read-write policy based on states. Gu et al. (2017) introduce an agent which learns to make decisions on when to translate from the interaction with a pre-trained offline neural machine translation model. Luo et al. (2017) used continuous rewards policy gradient for online alignments for speech recognition. Lawson et al. (2018) proposed a hard alignment with variational inference for online decoding. Alinejad et al. (2018) propose a new operation "predict" which predicts future source tokens. Zheng et al. (2019b) introduce a restricted dynamic oracle and restricted imitation learning for simultaneous translation. Zheng et al. (2019a) train the agent with an action sequence from labels that are generated based on the rank of the gold target word given partial input. Models from the last category leverage monotonic attention and replace the softmax attention with an expected attention calculated from a stepwise Bernoulli selection probability. Raffel et al. (2017) first introduce the concept of monotonic attention for online linear time decoding, where the attention only attends to one encoder state at a time. Chiu & Raffel (2018) extended that work to let the model attend to a chunk of encoder state. Arivazhagan et al. (2019) also make use of the monotonic attention but introduce an infinite lookback to improve the translation quality.

## 6  CONCLUSION

In this paper, we propose two variants of the monotonic multihead attention model for simultaneous machine translation. By introducing two new targeted loss terms which allow us to control both latency and attention span, we are able to leverage the power of the Transformer architecture to achieve better quality-latency trade-offs than the previous state-of-the-art model. We also present detailed ablation studies demonstrating the efficacy and rationale of our approach. By introducing these stronger simultaneous sequence-to-sequence models, we hope to facilitate important applications, such as high-quality real-time interpretation between human speakers.

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

## A  APPENDIX

### A.1  HYPERPARAMETERS

The hyperparameters we used for offline and monotonic transformer models are defined in Table 4.

### A.2  LATENCY METRICS DEFINITIONS

Given the delays $\mathbf{g} = \{g_1, g_2, ..., g_{|\boldsymbol{y}|}\}$ of generating each target token, AP, AL and DAL are defined in Table 5.

| Hyperparameter | WMT15 German-English | IWSLT English-Vietnamese |
|---|---|---|
| encoder embed dim | 1024 | 512 |
| encoder ffn embed dim | 4096 | 1024 |
| encoder attention heads | 16 | 4 |
| encoder layers | 6 | |
| decoder embed dim | 1024 | 512 |
| decoder ffn embed dim | 4096 | 1024 |
| decoder attention heads | 16 | 4 |
| decoder layers | 6 | |
| dropout | 0.3 | |
| optimizer | adam | |
| adam-$\beta$ | $(0.9, 0.98)$ | |
| clip-norm | 0.0 | |
| lr | 0.0005 | |
| lr scheduler | inverse sqrt | |
| warmup-updates | 4000 | |
| warmup-init-lr | 1e-07 | |
| label-smoothing | 0.1 | |
| max tokens | $3584 \times 8 \times 8 \times 2$ | 16000 |

**Table 4:** Offline and monotonic models hyperparameters.

| Latency Metric | Calculation |
|---|---|
| Average Proportion | $$\frac{1}{|\boldsymbol{x}||\boldsymbol{y}|} \sum_{i=1}^{|\boldsymbol{y}|} g_i$$ |
| Average Lagging | $$\frac{1}{\tau} \sum_{i=1}^{\tau} g_i - \frac{i-1}{|\boldsymbol{y}|/|\boldsymbol{x}|}$$ where $\tau = \arg\max_i (g_i = |\boldsymbol{x}|)$ |
| Differentiable Average Lagging | $$\frac{1}{|\boldsymbol{y}|} \sum_{i=1}^{|\boldsymbol{y}|} g_i' - \frac{i-1}{|\boldsymbol{y}|/|\boldsymbol{x}|}$$ where $g_i' = \begin{cases} g_i & i = 0 \\ \max(g_i, g_{i-1}' + \frac{|\boldsymbol{y}|}{|\boldsymbol{x}|}) & i < 0 \end{cases}$ |

**Table 5:** The calculation of latency metrics, given source $\boldsymbol{x}$, target $\boldsymbol{y}$ and delays $\boldsymbol{g}$

## A.3 DETAILED RESULTS

We provide the detailed results in Figure 2 as Table 6 and Table 7.

## A.4 THRESHOLD OF READING ACTION

We explore a simple method that can adjust system's latency at inference time without training new models. In Algorithm 1 line 8, 0.5 was used as an threshold. One can set different threshold $p$ during the inference time to control the latency. We run the pilot experiments on IWSLT15 En-Vi dataset and the results are shown as Table 8. Although this method doesn't require training new model, it dramatically hurts the translation quality.

|  | BLEU | AP | AL | DAL |
|---|---|---|---|---|
| $\lambda_{avg}$ | MMA-IL | | | |
| 0.05 | 30.7 | 0.78 | 10.91 | 12.64 |
| 0.1 | 30.5 | 0.70 | 7.42 | 8.82 |
| 0.2 | 30.1 | 0.63 | 5.17 | 6.41 |
| 0.3 | 30.3 | 0.60 | 4.18 | 5.35 |
| 0.4 | 29.2 | 0.59 | 3.75 | 4.90 |
| 0.5 | 26.7 | 0.59 | 3.69 | 4.83 |
| 0.75 | 25.5 | 0.58 | 3.40 | 4.46 |
| 1.0 | 25.1 | 0.56 | 3.00 | 4.03 |
| $\lambda_{var}$ | MMA-H | | | |
| 0.1 | 28.5 | 0.74 | 8.94 | 10.83 |
| 0.2 | 28.9 | 0.69 | 6.82 | 8.622 |
| 0.3 | 29.2 | 0.64 | 5.45 | 7.03 |
| 0.4 | 28.5 | 0.59 | 3.90 | 5.21 |
| 0.5 | 28.5 | 0.59 | 3.88 | 5.19 |
| 0.6 | 29.6 | 0.56 | 3.13 | 4.32 |
| 0.7 | 29.1 | 0.56 | 2.93 | 4.10 |

**Table 6:** Detailed results for MMA-H and MMA-IL on WMT15 DeEn

|  | BLEU | AP | AL | DAL |
|---|---|---|---|---|
| $\lambda$ | MILk | | | |
| 0.1 | 24.62 | 0.71 | 5.93 | 7.19 |
| 0.2 | 24.68 | 0.67 | 4.90 | 5.97 |
| 0.3 | 24.31 | 0.65 | 4.45 | 5.43 |
| 0.4 | 23.73 | 0.64 | 4.28 | 5.24 |
| $\lambda_{avg}$ | MMA-IL | | | |
| 0.02 | 28.28 | 0.76 | 7.09 | 8.29 |
| 0.04 | 28.33 | 0.70 | 5.44 | 6.57 |
| 0.1 | 28.42 | 0.67 | 4.63 | 5.65 |
| 0.2 | 28.47 | 0.63 | 3.57 | 4.44 |
| 0.3 | 27.9 | 0.59 | 2.98 | 3.81 |
| 0.4 | 27.73 | 0.58 | 2.68 | 3.46 |
| $\lambda_{var}$ | MMA-H | | | |
| 0.02 | 27.26 | 0.77 | 7.52 | 8.71 |
| 0.1 | 27.68 | 0.69 | 5.22 | 6.31 |
| 0.2 | 28.06 | 0.63 | 3.81 | 4.84 |
| 0.4 | 27.79 | 0.62 | 3.57 | 4.59 |
| 0.8 | 27.95 | 0.60 | 3.22 | 4.19 |

**Table 7:** Detailed results for MILk, MMA-H and MMA-IL on IWSLT15 En-Vi

## A.5 AVERAGE LOSS FOR MMA-H

We explore applying a simple average instead of a weighted average loss to MMA-H. The results are shown in Figure 7 and Table 9. We find that even with very large weights, we are unable to reduce the overall latency. In addition, we find that the weighted average loss severely affects the translation quality negatively. On the other hand, the divergence loss we propose in Equation 16 can efficiently reduce the latency while retaining relatively good translation quality for MMA-H models.

| | Reading Threshold | | | | Weighted Average Latency Loss | | | |
|---|---|---|---|---|---|---|---|---|
| $p$ | BLEU | AP | AL | DAL | $L_{avg}$ | BLEU | AP | AL | DAL |
| 0.5 | 25.5 | 0.792387 | 7.13673 | 8.27187 | 0.02 | 25.5 | 0.792387 | 7.13673 | 8.27187 |
| 0.4 | 25.25 | 0.73749 | 5.72003 | 6.85812 | 0.04 | 25.68 | 0.728107 | 5.52856 | 6.61744 |
| 0.3 | 23.06 | 0.697398 | 4.88087 | 6.03342 | 0.3 | 24.9 | 0.602703 | 2.90054 | 3.68039 |
| 0.2 | 18.37 | 0.678298 | 4.71099 | 5.94636 | 0.2 | 25.3 | 0.636914 | 3.54577 | 4.38623 |
| 0.1 | 8.73 | 0.696452 | 5.5225 | 7.20439 | 0.1 | 25.48 | 0.684424 | 4.57901 | 5.54102 |

**Table 8:** Comparison between setting threshold for reading action and weighted average latency loss.

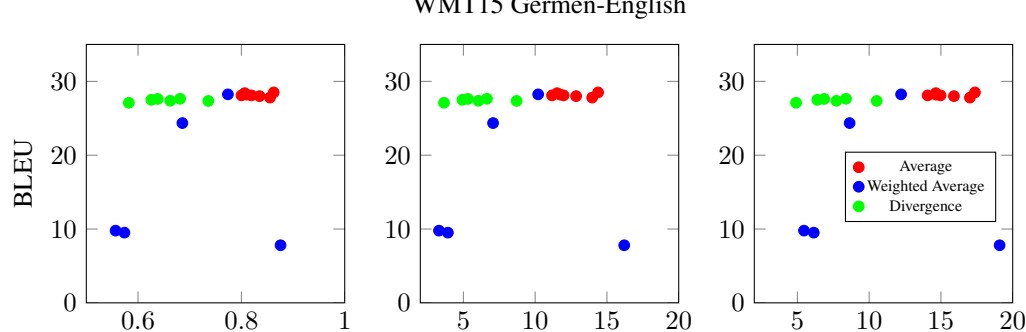

**Figure 7:** Effect average loss, weighted average loss and variance loss on MMA-H on WMT15 DeEn development set.

| | BLEU | AP | AL | DAL |
|---|---|---|---|---|
| $\lambda_{avg}$ | | Average | | |
| 0.05 | 28.5 | 0.862581 | 14.3847 | 17.3702 |
| 0.1 | 27.8 | 0.855435 | 13.974 | 17.02 |
| 0.2 | 28 | 0.835324 | 12.8531 | 15.908 |
| 0.4 | 28.1 | 0.819408 | 11.9816 | 14.9763 |
| 1.0 | 28.2 | 0.810609 | 11.7528 | 14.6695 |
| 2.0 | 28.1 | 0.800258 | 11.1763 | 14.0761 |
| 8.0 | 28.4 | 0.806439 | 11.5289 | 14.6431 |
| $\lambda_{avg}$ | | Weighted Average | | |
| 0.02 | 28.24 | 0.773922 | 10.2109 | 12.2274 |
| 0.04 | 24.35 | 0.685834 | 7.06716 | 8.64069 |
| 0.06 | 7.80 | 0.875825 | 16.2046 | 19.0892 |
| 0.08 | 9.51 | 0.57372 | 3.92011 | 6.1421 |
| 0.1 | 9.78 | 0.556585 | 3.3007 | 5.46142 |
| $\lambda_{var}$ | | Divergence | | |
| 0.1 | 27.35 | 0.736025 | 8.70968 | 10.5253 |
| 0.2 | 27.64 | 0.681491 | 6.63914 | 8.3856 |
| 0.3 | 27.37 | 0.6623 | 6.04902 | 7.71922 |
| 0.4 | 27.62 | 0.638188 | 5.31672 | 6.86834 |
| 0.5 | 27.50 | 0.625759 | 4.93044 | 6.38998 |
| 1.0 | 27.1 | 0.582194 | 3.64864 | 4.90997 |

**Table 9:** Detailed numbers on average loss, weighted average loss and head divergence loss on WMT15 De-En development set

