# OpenReview forum: "Monotonic Multihead Attention"
_ICLR.cc/2020/Conference — Accept (Poster)_

### Official Review · AnonReviewer2 · 2019-10-23
**Official Blind Review #2**

**Rating:** 6

**Review:**

This paper proposes a fully transformer-based monotonic attention framework that extends the idea of MILK. Though the idea of monotonic multi-head attention sounds interesting, I still have some questions below:

About the method:
   1. Is that possible that the MMA would have worse latency than MILK since all the attention heads need to agree to write while MILK only has one attention head?
   2. Is there any attention order between different attention head?
   3. I think the MMA only could control the latency during training time, which would produce different models with different latency. Is there any way that enables MMA to control the latency during inference time? Can we change the latency for on given model by tuning the requirements mentioned in (1)?

About the experiments:
    1. Do you have any explanation of why both MMA-H and MMA-IL have better BLEU when AL is small? The results in fig 2 seem counterintuitive.
    2. I suggest the authors do more analysis of the difference between different attention heads to prove the effectiveness of MMA.
    3. For the left two figures in fig 4, which one is the baseline, and which one is the proposed model?

I also suggest the authors present more real sample analysis and discussions about the experiments.

**Experience Assessment:**

I have published one or two papers in this area.

**Review Assessment: Checking Correctness Of Derivations And Theory:**

I assessed the sensibility of the derivations and theory.

**Review Assessment: Checking Correctness Of Experiments:**

I carefully checked the experiments.

**Review Assessment: Thoroughness In Paper Reading:**

I read the paper at least twice and used my best judgement in assessing the paper.

---

> ### Author Response · Authors · 2019-11-13
> **Response to questions from AnonReviewer2**
>
> About the method:
> 1. Thank you for the great question. The overall latency are mainly introduced by the fastest head. It is possible when there is a small weight on the latency regularization term. By using proper regularization weight for the latency control method we proposed, we can “slow down” the faster heads so even though all heads need to agree to write, we can still achieve a low latency. Moreover, both latency regularization methods punish more on the faster head.
>
> 2. Thank you for the great question. We carried out an analysis on attention head rank during the inference time. Figure 6 shows that some heads can run faster than others and it is more obvious in MMA-IL. It can also be observed from the case example in Figure 5b. We also find that the higher layers are likely to move faster than lower layers, especially in MMA-IL.
>
> 3.This is a very good question. We have carried out one simple experiment that only change the inference setting, which is shown in Appendix A.4. However, we observe that this method failed to produce high quality translations because there is a mismatch between training and inference time. This training-inference mismatch was also observed in Ma et al. (2019), where they report a significant difference with wait-k and test wait-k.
>
> Finally, would you be able to clarify what you mean by “tuning the requirements mentioned in (1)” ?
>
> About the experiments:
> 1. This is a very good question. The MMA-IL or MILk model, which has an infinite look back attention mechanism, will have more information when latency increases because of the larger context coverage on source side. An extreme case is that when the model reach the maximum latency (AP=1), the MMA-IL or MILk will behave like the full offline models. However, for MMA-H, because of the hard alignment, a larger latency does not necessarily mean an increase in source information available to the model. In fact, the large latency was introduced by outlier attention heads, which read the entire source sentence and point to the end of the source sentence. These outlier heads not only increase the latency but they also do not provide useful information to the model as they are pointing to the end of sentence token, which is not very informative. We introduce the attention variance loss to eliminate the outliers, as such a loss makes the attention heads focus on the current context for translating the new target token. This is why when the latency decreases (fewer outliers), the MMA-H model has an increase on translation quality.
>
> 2. Thank you for the suggestion. We have added a visualization of a real sample as well as an analysis on how heads behave at different layers.
>
> 3. Thank you for pointing this out. We have updated the figure to make it clear. Each subplot now has a title (“Offline Model”, “MMA-H”, “MMA-H (latency)”, “MMA-IL”, “MMA-IL (latency)”).
>
> Thank you for the suggestion. We have added a visualization of a real sample as well as an analysis on how heads behave at different layers.

---

### Official Review · AnonReviewer1 · 2019-10-24
**Official Blind Review #1**

**Rating:** 8

**Review:**

Summary: This paper applies monotonic attention to the multiheaded (self-attention) mechanisms used in a Transformer. It also proposes a few new losses which encourage low-latency alignments. Experiments are carried out on WMT EnDe and IWSLT EnVi translation, with evaluation using BLEU and latency-related metrics.

Review: Applying monotonic attention mechanisms to the Transformer architecture is an obvious and necessary idea, and as such this paper constitutes an important contribution. The application of monotonic attention to the Transformer is as I would have expected. The latency-reduction losses are novel, however. I think there are various things the authors could do to clarify their results as well as the presentation of their methods, which I discuss below. Overall, I think this is a solid accept, especially with some improvements to the presentation.

Specific comments & suggestions:
- There is a typo in the abstract: "multiple attentions heads"
- "the denominator in Equation 5 is clamped into a range of (\eps, 1]" Technically this doesn't need to be an open range on the left.
- "which allows the decoder to apply softmax attention over a chunk (subsequence of encoder positions)." It may be clearer if you just say "which allows the decoder to apply to a fixed-length subsequence of encoder states preceding $t_i$".
- I think more could be done to distinguish the behavior of the encoder self-attention, the encoder-decoder attention, and the decoder self-attention. You write "The model will write a new target token only after all the attentions have decided to write" but also "Some heads read new inputs, while the others can stay in the past to retain the source history information." If an attention head is "staying in the past", then the model will not be able to write a new target token. I think in one of these cases you are referring to (encoder) self-attention and in the other you are referring to decoder attention. Please clarify.
- "Although MMA-H is not quite yet streaming capable since both the encoder and decoder self-attention have an infinite lookback, that model represents a good step in that direction." I think you should distinguish between online and streaming translation. It sounds like when you say streaming you mean that the utterances may continue arbitrarily, so infinite lookback is impractical. However, one could truncate the source sequence whenever the decoder outputs an end-of-sentence token, or something. I'm not sure people usually make a strong distinction here.
- For completeness it would be useful to include at least wait-k, and potentially a line corresponding to offline attention performance, in the plots in Figure 2.
- Why don't you include MMA-IL in WMT'15 EnDe? (figure 2)
- Are you copying the results from (Arivazhagan et al., 2019) or did you reimplement MILk in your codebase?
- The results in Figure 2 could be made much more informative if there was some way of communicating the multiplier (the "lambdas") of the different latency losses for each of the different dots on the plot. This would make it much more obvious how important these losses are and how the effect the quality/latency trade-off.
- There are some additional possible references for online seq2seq, like CTC, the Neural Transducer, "Learning Hard Alignments with Variational Inference", "Learning online alignments with continuous rewards policy gradient”, etc.

**Experience Assessment:**

I have published in this field for several years.

**Review Assessment: Checking Correctness Of Derivations And Theory:**

I assessed the sensibility of the derivations and theory.

**Review Assessment: Checking Correctness Of Experiments:**

I assessed the sensibility of the experiments.

**Review Assessment: Thoroughness In Paper Reading:**

I read the paper thoroughly.

---

> ### Author Response · Authors · 2019-11-13
> **Response to specific comments & suggestion from AnonReviewer1**
>
> - There is a typo in the abstract: "multiple attentions heads"
>
> Thank you, this was corrected.
>
> - "the denominator in Equation 5 is clamped into a range of (\eps, 1]" Technically this doesn't need to be an open range on the left.
>
> Thank you for the comment, we have updated this to “[” after checking the implementation.
>
> - "which allows the decoder to apply softmax attention over a chunk (subsequence of encoder positions)." It may be clearer if you just say "which allows the decoder to apply to a fixed-length subsequence of encoder states preceding ".
>
> Thank you, the text has been edited accordingly.
>
> - I think more could be done to distinguish the behavior of the encoder self-attention, the encoder-decoder attention, and the decoder self-attention. You write "The model will write a new target token only after all the attentions have decided to write" but also "Some heads read new inputs, while the others can stay in the past to retain the source history information." If an attention head is "staying in the past", then the model will not be able to write a new target token. I think in one of these cases you are referring to (encoder) self-attention and in the other you are referring to decoder attention. Please clarify.
>
> We have provided more details the algorithm 1 on how to update encoder states during the inference. However, both sentences are talking about encoder-decoder attention.
>
> "The model will write a new target token only after all the attentions have decided to write"  indicate when to start the writing process, which is line 19 in algorithm 1.
>
> “Some heads read new inputs, while others can stay in the past to retain the source history information.” means that different heads have different speeds when reading the source tokens. If one attention head decides to write, as it shows in line 8-10, the head will stay on this location and wait for other heads finish reading. If an attention head can be "staying in the past" as long as j is smaller than t_max, which is the actually number of tokens read by model.
>
> - "Although MMA-H is not quite yet streaming capable since both the encoder and decoder self-attention have an infinite lookback, that model represents a good step in that direction." I think you should distinguish between online and streaming translation. It sounds like when you say streaming you mean that the utterances may continue arbitrarily, so infinite lookback is impractical. However, one could truncate the source sequence whenever the decoder outputs an end-of-sentence token, or something. I'm not sure people usually make a strong distinction here.
>
> What we mean here is that hard attention can handle an input stream natively, without having to resort to segmenting the input. However, this is a good point that in practice, both our model and MILk are able to handle arbitrarily long input, provided that we segment the input. We updated the explanation accordingly.
>
> - For completeness it would be useful to include at least wait-k, and potentially a line corresponding to offline attention performance, in the plots in Figure 2.
>
> We updated the paper with the offline baseline in Figure 2. We didn’t include wait-k (Ma et al. 2019) and reinforcement learning models (Gu et al. 2017) because they underperform the MILk model (Arivazhagan et al. (2019)).
>
> - Why don't you include MMA-IL in WMT'15 EnDe? (figure 2)
>
> Thank you for the suggestion. The updated paper now includes the MMA-IL results on the WMT15 De-En dataset.
>
> - Are you copying the results from (Arivazhagan et al., 2019) or did you reimplement MILk in your codebase?
>
> For the IWSL15 En-Vi dataset, we implemented monotonic infinite lookback models based on LSTM, following the settings from Colin Raffel et al (2017) and Chiu and Raffel (2018). For WMT15 De-En, we copied the numbers from (Arivazhagan et al., 2019) because they are using a RNMT+ implementation.
>
> - The results in Figure 2 could be made much more informative if there was some way of communicating the multiplier (the "lambdas") of the different latency losses for each of the different dots on the plot. This would make it much more obvious how important these losses are and how the effect the quality/latency trade-off.
>
> Thank you for the suggestion, we updated the detailed results in the appendix.
>
> - There are some additional possible references for online seq2seq, like CTC, the Neural Transducer, "Learning Hard Alignments with Variational Inference", "Learning online alignments with continuous rewards policy gradient”, etc.
>
> Thank you so much suggestions, we updated the related work.

---

> > ### Comment · AnonReviewer1 · 2019-11-15
> > **Response**
> >
> > Thanks for your clarifications.

---

### Official Review · AnonReviewer4 · 2019-10-31
**Official Blind Review #4**

**Rating:** 6

**Review:**

The paper proposes an approach for simultaneous neural machine translation. While prior works deal with recurrent models, the authors adopt previous approaches for Transformer. Specifically, for decoder-encoder attention they introduce Monotonic Multihead Attention (MMA) designed to deal with several attention heads. Each attention head operates as a separate monotonic attention head similar to the ones from previous works by Raffel et al. (2017) and Arivazhagan et al. (2019). MMA has two versions: MMA-H (hard), where each head attends only to one token, and MMA-IL (infinite lookback), where each head attends to all tokens up to a selected position.

The main novelty of the work in the way of dealing with multiple heads: (i) average over heads latency instead of single-head latency, (ii) penalty which encourages different heads point to similar positions (L_var).

Experimental results on two translation tasks (IWSLT En-Vi and WMT De-En) show better quality-latency trade-offs compared to the recurrent analogs. Experiments with different number of attention heads would be helpful, but in the current state are rather confusing (see the comments below).


I can not recommend accepting this paper due to the two main reasons.

1) The proposed solution lacks novelty.
MMA-H and MMA-IL attentions, introduced in this work, are straightforward applications of previous works by Raffel et al. (2017) and Arivazhagan et al. (2019) respectively to each of the heads. Novelty is in the way of dealing with multiple heads: (i) average over heads latency instead of single-head latency, (ii) penalty which encourages different heads point to similar positions. However, these alone are unlikely to make a large impact.

2) The results are experimentally weak, evaluation is questionable.
The current baseline is the recurrent model, but since the main contribution of this work is in how to deal with several heads the proper baselines would be (i) the same, but with single-head attention, (ii) the same, but without L_var (summing latency penalty over heads is straightforward). However, these baselines are absent and the improvement is likely to be due to the replacement of RNN with the Transformer - this would be a limited contribution.



Other comments on the experiments.

1) MMA-IL was tested only on one small dataset (IWSLT En-Vi), MMA-H only on 2 datasets - more experiments would help.

2) Figure 2 shows that when increasing lagging, performance first improves, then drops. While the former is expected, the latter is not: one would expect a model to reach the performance of its offline version when given maximum latency, but not to drop. The MILK model (Arivazhagan et al., 2019) behaves as expected, but the proposed MMA does not: does this mean that there is some problem with the model? An explanation of this behavior would be helpful.

3) Number of heads, Figure 4: the results are supposed to show that performance improves when increasing the number of heads, but the figure shows the opposite. If the decoder has more than one layer (which Transformer usually has), for 4 heads perform worse than one, 8 heads (which the standard Transformer setting) - not better than one, and we get the improvement only when using 16 heads with 6 layers. Could you elaborate on this? The text says “quality generally tends to improve with more layers and more heads”.

4) I can see the motivation for using the L_var loss term in this setting, but forcing heads in MMA-H attend to similar positions seems counterintuitive: there is evidence that making attention heads less correlated improves quality (see for example Li et al, EMNLP 2018 https://www.aclweb.org/anthology/D18-1317/), but L_var may end up doing the opposite. How different is the behavior of the learned heads? Figure 3 shows the average attention span, but this does not tell how many of the heads are doing different things. Some analysis would be nice.


Comments on the presentation.

1) It would be better to mention that encoder self-attention is limited only to previous positions earlier in the text and more prominently (now it’s on the bottom of the sixth page). A good place would be in Section 2.2 when talking about different attention types in the Transformer and modifications required for simultaneous translation.

2) Figure 2: it would be helpful to see offline model performance on these pictures (e.g., a horizontal line showing the BLEU scores). One of the main points of these experiments is to see how much each model drops compared to its offline version.

3) Figure 4: it is not clear from the figure which models it corresponds (for example, what is the difference between the first and the second figures: language pairs? models?)

4) Table 2 shows the BLEU scores for Transformer with unidirectional encoder and greedy decoding. It would be helpful to see also the BLEU scores of Transformer in the standard setting (full attention in the encoder, beam search inference). For now, it is not clear how much one loses by replacing the encoder to unidirectional: if much, then probably it makes sense to work also on the encoder (read and encode source sentence with a latency).

Language and typos.
The text is quite raw and could use an extra work; a lot of typos starting from the abstract (e.g., “...for multiple attentions heads”).


**Experience Assessment:**

I have read many papers in this area.

**Review Assessment: Checking Correctness Of Derivations And Theory:**

I assessed the sensibility of the derivations and theory.

**Review Assessment: Checking Correctness Of Experiments:**

I assessed the sensibility of the experiments.

**Review Assessment: Thoroughness In Paper Reading:**

I read the paper at least twice and used my best judgement in assessing the paper.

---

> ### Author Response · Authors · 2019-11-13
> **Response to the two reasons not accepting this paper from AnonReviewer4**
>
> 1) We thank the reviewer for pointing out prior work on monotonic attention based models and latency augmented training for simultaneous machine translation. We’re convinced that our work has significant differences with prior work (Raffel et al. 2017, Arivazhagan et al. 2019):
>
> a.To the best of our knowledge, we are the first to enable transformers to have a learnable policy for simultaneous translation. Although monotonic attention and monotonic infinite lookback attention were introduced before this work, it is non-trivial to make them compatible with multihead attention. In fact, Arivazhagan et al. (2019) do not use multihead attention even though the RNMT+ model originally supports multihead attention (https://arxiv.org/abs/1804.09849).
>
> b.We introduce two novel models, MMA-H and MMA-IL. MMA-H is designed with the eventual goal of naturally handling arbitrarily long input while the goal of MMA-IL is to retain as much quality as possible with respect to an offline model. These two models require two carefully designed losses, described next, in order to be effective.
>
> c.The variance loss we introduce required preliminary experimentation and careful design. This loss encourages different heads to point to positions close to each other. In preliminary experiments, we found that some heads tend to skip every token and move to the end of sentence while some heads tend to stay at the beginning of the sentence. In the case of the MMA-H model, the fast outliers heads increase latency but do not improve quality since they point to an uninformative token (end of sentence) while the slow outlier heads prevent the model to deal with arbitrary long input in a natural way. Our regularization was designed specifically for hard monotonic multihead attention and, to the best of our knowledge, we are the first to propose this idea for latency control.
>
> d.Finally, we believe this work will have an impact on the community for three reasons:
> i. We reach state-of-the-art performance for simultaneous machine translation.
> ii. We are the first to combine the transformer with monotonic attention to introduce a learnable policy transformer model which outperforms the wait-k model (which is also transformer-based).
> iii. We will publish the code to contribute to the community of simultaneous translation. Note that very little prior work did so.
>
>
> 2) Thank you for the suggestions.
>
> We picked (Arivazhagan et al. 2019) as our baseline since this represents the current state-of-the-art for simultaneous machine translation. Note that while the monotonic infinite lookback attention (MILk) (Arivazhagan et al. 2019) is an RNN-based model, it still outperforms previous transformer-based approaches such as the wait-k model (Ma et al. 2019, https://www.aclweb.org/anthology/P19-1289/).
>
> Regarding the first proposed baseline (“the same, but with single-head attention”), note that as soon as we use multiple layers, our proposed algorithm/model is necessary since there are multiple encoder-decoder attention layers. The only situation for the transformer that does not require a modification of the previously proposed MILk model is when there is single-head attention *and* only one layer. We have also updated Figure 4 with additional experimentation for single-head attention and the MMA-IL model.
>
> We corrected the text of the paper which said “For MMA-IL model, we used both loss terms;”, sorry for the mistake. For that model, we were actually only using the proposed weighted average loss. In summary, we are only using L_var for MMA-H and only using L_avg for MMA-IL. Note that the weighted average loss we introduce is different than the straightforward average loss proposed by the reviewer. The method we proposed automatically assigns more weights on faster heads and reduces overall latency. It also moderately regularizes the slower heads to prevent them for being outliers later. We are doing an additional experiment with this simple average loss.
>
> One of the motivation to use MMA-H is to not have heads which are particularly slow and eventually enable streaming in a natural way. In preliminary experiments, we observe that for MMA-H models, some heads always stays at the beginning of the sentence. This is why we introduced L_var, in order to have different heads stay together. As you can see in Figure 3, L_var is effective from that perspective. We are also doing additional experiments with the simple average loss for MMA-H proposed by the reviewer.

---

> > ### Comment · AnonReviewer2 · 2019-11-14
> > **papers with flexible policy**
> >
> > Some extra information about your statement "a. To the best of our knowledge, we are the first to enable transformers to have a learnable policy for simultaneous translation."
> >
> > At least I know two papers that are using a learnable policy with pure transformer models.
> >
> > [1] "Simpler and Faster Learning of Adaptive Policies for Simultaneous Translation" in EMNLP 2019
> >
> > [2]"Simultaneous Translation with Flexible Policy via Restricted Imitation Learning" in ACL 2019
> >
> > could you also comment on the difference between those papers as well?

---

> > > ### Author Response · Authors · 2019-11-15
> > > **Response to papers with flexible policy**
> > >
> > > Thank you for pointing out these related papers. We will also cite these two additional references before the rebuttal deadline.
> > >
> > > In the second paper (Zheng et al 2019, ACL), the authors propose to use a restricted dynamic oracle, which is controlled by a conservative and an aggressive bound, in order to train their model. The authors also introduced a delay token to give the model information about the READ action. The reviewer is correct in pointing out that this paper is an example of using the Transformer together with a learnable policy.
> > >
> > > Differences between (Zheng et al 2019, ACL) and our work include the following:
> > > 1. In our work, we used a closed form solution of expected alignment for training, while (Zheng et al 2019, ACL) introduced a sampling-based restricted imitation learning.
> > > 2. (Zheng et al 2019, ACL) modify the model by introducing a special delay token in the target vocabulary while our model introduces read/write probabilities for each attention head.
> > > 3. In our work, the actions are fully decided by the model, while in (Zheng et al 2019, ACL) hard constraints were applied at test time to control the latency.
> > > 4. Unfortunately, we are unable to compare our work to (Zheng et al 2019, ACL) within the rebuttal timeframe, especially since the experiments involve the NIST corpus, which is not completely open (it is open to the participants of the OpenMT evaluations).
> > >
> > > The first paper (Zheng et al 2019, EMNLP) was just published in EMNLP 2019, which was after the submission deadline for ICLR 2020, and the arxiv version was posted on September 4 2019, which is only 20 days before the ICLR submission deadline so it is effectively concurrent work. Thanks for bringing it to our attention. This paper indeed proposed a trainable policy model for Transformer. First, a sequence of actions were generated based on the rank of the gold target word given partial input given by an offline model. The generated action sequences were then used as labels in order to train an agent in a supervised fashion. At inference time, an offline model was used where actions came from the trained agent. The latency can be controlled by the threshold when generating the action sequence.
> > >
> > > Differences between our work and (Zheng et al 2019, EMNLP) include the following:
> > > 1. In our work, the policy and model were trained jointly while in (Zheng et al 2019, EMNLP), the agent was trained separately from the translation model. Since the translation model is not adapted to different agents, this can potentially introduce a mismatch between training and inference.
> > > 2. In our work, we trained different translation models for different latency requirements, while (Zheng et al 2019, EMNLP) use one model for all latency regimes, this could also introduce a mismatch and thus hurt the performance.
> > > 3. In our work, we didn’t use explicit labels to train the policy, while (Zheng et al 2019, EMNLP) used generated labels to train the policy and the generating actions are controlled by a hyperparameter \rho.
> > > 4. Our work obtains state-of-the-art performance, while the models in (Zheng et al 2019, EMNLP) underperformed MILk.

---

> ### Author Response · Authors · 2019-11-13
> **Response to the comments on the experiments**
>
> 1)Thank you for the suggestion. The updated paper now includes the MMA-IL results on the WMT15 De-En dataset, which are also positive. We believe that 2 datasets is appropriate to demonstrate the effectiveness of our method, as was done in prior work. However, we will conduct follow up experimentation on WMT14 English-French in order to have an additional comparison datapoint with Arivazhagan et al. (2019) for the camera-ready version, should this paper be accepted.
>
> 2) This is a very good question. The MMA-IL or MILk model, which has an infinite look back attention mechanism, will have more information when latency increases because of the larger context coverage on source side. An extreme case is that when the model reach the maximum latency (AP=1), the MMA-IL or MILk will behave like the full offline models. However, for MMA-H, because of the hard alignment, a larger latency does not necessarily mean an increase in source information available to the model. In fact, the large latency is introduced by outlier attention heads, which read the entire source sentence and point to the end of the source sentence. These outlier heads not only increase the latency but they also do not provide useful information to the model as they are pointing to the end of sentence token, which is not very informative. We introduce the attention variance loss to eliminate the outliers, as such a loss makes the attention heads focus on the current context for translating the new target token. This is why when the latency decreases (fewer outliers), the MMA-H model has an increase on translation quality.
>
> 3)Thank you for pointing this out. The purpose of this experiment is to motivate the introduction of the MMA model. The Transformer model usually has multiple encoder-decoder attention heads, either because it has multiple layers (even with single-head attention, there is more than one head: one head per layer), or because it uses multihead attention. We have updated Figure 4 to include the single-head attention case and we also included the results to include the MMA-IL model. The updated figure demonstrates that multiple heads (either coming from multiple layers or from using multihead attention) is beneficial for performance up to a certain point. We also observe that with one layer, increasing the number of heads is beneficial. With more layers, for the baseline and MMA-IL, increasing the number of heads is beneficial up to a certain point, then performance plateaus or degrades. For MMA-H, using more than one head degrades performance initially, then the performance starts recovering. In any case, the single layer/single head case underperforms all other settings, which motivates the introduction of our model.
>
> 4) Thank you for the great question. The setting in our paper is different from Li et al (2018), especially for MMA-H. The attention is monotonic and hard-aligned to source states in MMA-H, whereas Li et al (2018) uses the full softmax attention. Furthermore, the disagreement of attention in Li et al (2018) is different from the disagreement of attention head positions in our work. In Li et al (2018), disagreement of attention means disagreement of softmax distributions; in our work, the disagreement means the disagreement of attention head position during inference time. These two are not necessarily correlated. When we eliminate the outliers, we find that MMA-H works better because the hard aligned attention heads can focus more on the current source context for translating a new target word. We also add visualization and rank analysis in the paper.

---

> > ### Author Response · Authors · 2019-11-15
> > **Additional response to question 4 for experiments**
> >
> > We updated the results of average latency regularization for MMA-H in the appendix. We observed than neither the simple average nor the weighted average regularization work on MMA-H models. Even with large weights, we are unable to reduce the latency, while the weighted average severely hurts the translation quality.
> >
> > For MMA-IL, we also find that we are unable to reduce the latency with the range of weights explored. We are investigating the effect of larger weights for MMA-IL and the simple average loss. For reference, here are the numbers obtained so far:
> >
> > Weight | BLEU |AP | AL | DAL
> > 0.05 | 29.9 | 0.98 | 24.50 | 26.24
> > 0.1 | 30.0 | 0.95 | 21.16 | 24.01
> > 0.2 | 29.8 | 0.94 | 20.20 | 23.09
> > 0.4 | 29.9| 0.92 | 18.80 | 22.05
> > 1 |28.2 | 0.81 | 11.75 | 14.67

---

> ### Author Response · Authors · 2019-11-13
> **Response to the comments on the presentation**
>
> 1) Thank you for pointing this out. We have updated the paper accordingly and added this information in Section 2.2.
>
> 2) Thank you for the advice, we have updated the offline models in the figures.
>
> 3) Thank you for pointing this out. We have updated the figure to make it clear. Each subplot now has a title (“Offline Model”, “MMA-H”, “MMA-H (latency)”, “MMA-IL”, “MMA-IL (latency)”).
>
> 4) We have updated the results of unidirectional and bidirectional offline model with greedy and beam search. With beam search, the performance is not affected by a unidirectional encoder. Greedy decoding introduces a small drop of 0.4 BLEU and the unidirectional encoder further gives a 0.3 BLEU drop.
>
> Thank you for the comment. We proofread and edited accordingly.

---

> ### Public Comment · ~Gladis_Ne_Limes1 · 2023-09-12
> **rre**
>
> EHR software deals with highly sensitive patient information, making data security a top priority. Implementing robust security measures, including encryption, access controls, and audit trails https://mlsdev.com/blog/ehr-development, is essential to protect patient privacy and comply with regulations like HIPAA in the United States.

---

### Public Comment · ~Gladis_Ne_Limes1 · 2023-08-20
**re**

Navigating the world of web design has never been more enlightening https://claspo.io/blog/how-to-avoid-negative-impacts-from-annoying-pop-ups-on-the-website-laspo-tips-and-tools/, thanks to LASPO's guide on avoiding the pitfalls of pesky pop-ups. It's like having a seasoned mentor whispering sage advice in your ear while you build your digital domain. By embracing these tips and tools, you're not just sidestepping annoyances; you're fostering a harmonious user experience that leaves a positive imprint. In an era where user engagement is paramount, LASPO's insights are a compass pointing toward user-centric design and seamless interactions. So let's bid adieu to frustration and welcome a new era of web elegance!

---

### Decision · Program_Chairs · 2019-12-19

**Decision:**

Accept (Poster)

**Comment:**

This paper extends previous models for monotonic attention to the multi-head attention used in Transformers, yielding "Monotonic Multi-head Attention." The proposed method achieves better latency-quality tradeoffs in simultaneous MT tasks in two language pairs.

The proposed method is a relatively straightforward extension of the previous Hard and Infinite Lookback monotonic attention models. However, all reviewers seem to agree that this paper is a meaningful contribution to the task of simultaneously MT, and the revised version of the paper (along with the authors' comments) addressed most of the raised concerns.

Therefore, I propose acceptance of this paper.